# Stability Analysis of the Horseshoe Tunnel Face in Rock Masses

**DOI:** 10.3390/ma15124306

**Published:** 2022-06-17

**Authors:** Jun Liu, Qingsong Zhang, An Liu, Guanghui Chen

**Affiliations:** 1School of Civil Engineering, Shandong University, Jinan 250061, China; frogniuwa@163.com (J.L.); zhangqingsong@sdu.edu.cn (Q.Z.); 2Jiangxi Provincial Communications Investment Group Co., Ltd., Nanchang 330025, China; 3Jiangxi Transportation Consulting Co., Ltd., Nanchang 330008, China; havencher@163.com; 4School of Civil Engineering, Central South University, Changsha 410075, China

**Keywords:** horseshoe tunnel face, upper-bound limit analysis method, nonlinear Hoek–Brown failure criterion, limit support pressure, 3D failure surface

## Abstract

Accurately estimating the stability of horseshoe tunnel faces remains a challenge, especially when excavating in rock masses. This study aims to propose an analytical model to analyze the stability of the horseshoe tunnel face in rock masses. Based on discretization and “point-by-point” techniques, a rotational failure model for horseshoe tunnel faces is first proposed. Based on the proposed failure model, the upper-bound limit analysis method is then adopted to determine the limit support pressure of the tunnel face under the nonlinear Hoek–Brown failure criterion, and the calculated results are validated by comparisons with the numerical results. Finally, the effects of the rock properties on the limit support pressure and the 3D failure surface are discussed. The results show that (1) compared with the numerical simulation method, the proposed method is an efficient and accurate approach to evaluating the face stability of the horseshoe tunnel; (2) from the parametric analysis, it can be seen that the normalized limit support pressure of the tunnel face decreases with the increasing of geological strength index, *GSI*, Hoek–Brown coefficient, *m_i_*, and uniaxial compressive strength, *σ_ci_*, and with the decreasing of the disturbance coefficient of rock, *D_i_*; and (3) a larger 3D failure surface is associated with a high value of the normalized limit support pressure.

## 1. Introduction

With the development of shield technology, various tunnels, such as circular and non-circular tunnels (i.e., elliptical, rectangular, and horseshoe tunnels), are excavated in practical engineering. Compared with the circular tunnels, non-circular tunnels are more popular in practical engineering due to their higher section utilization and lower construction costs. To maintain the stability of the excavation face, the shield machine always provides a continuous supporting pressure at the tunnel face. A collapse failure will happen at the tunnel face when the supporting pressure is not sufficient to resist the movement of soil or rock towards the tunnel. Therefore, in the excavation of shield tunnels, one of the most important issues is to determine the required minimum supporting pressure to ensure the stability of the excavation face.

To solve this problem, the following three methods have been repeatedly adopted: (1) experimental tests [1,2]; (2) numerical simulations [3,4]; (3) analytical approaches [5,6,7,8,9,10]. Because, compared with experimental tests and numerical simulations, the analytical approaches are easier to implement and can provide a vehicle for engineers to directly design required minimum supporting pressures for the tunnel face [8], the analytical approaches, especially the upper-bound limit analysis approach, are the most popular methods to assess stability issues. Recently, many scholars have proposed various analytical models to investigate tunnel face stability [5,6,7,8]. For instance, considering the soil arching effect, Han et al. [5] proposed a failure model composed of five conical blocks to estimate the stability of circular tunnel faces. Based on the discretization technique, a rotational failure mechanism was proposed by Mollon et al. [6] to assess the face stability of circular tunnels. Among these analytical failure models, the rotational failure model proposed by Mollon et al. [6] is recognized as the most popular one due to the fact that the whole circular tunnel face is considered [7,8]. Since the study by Mollon et al. [6], the rotational failure model has been widely adopted by many researchers to estimate tunnel stability under various complex constraints [9,10,11,12,13]. It is worth noting that these studies are limited to circular tunnels and the linear Mohr–Coulomb failure criterion.

Additionally, some analytical failure models have been proposed to analyze the face stability of non-circular tunnels [14,15,16]. For example, Chen et al. [14] proposed an improved pyramid failure model to estimate stability of rectangular tunnel faces considering the soil-arching effect in sandy soils. In the framework of the upper-bound limit analysis method, Chen et al. [15] constructed a discrete failure mechanism to evaluate the stability of rectangular tunnel faces in nonhomogeneous soils. Based on the limit equilibrium method, Xie et al. [16] studied the stability of rectangular tunnel faces reinforced by umbrella pipes. Note that these analytical models are only focused on rectangular tunnel faces. Few studies have investigated the stability of horseshoe tunnel faces, except Pan and Dias [17]. In the study by Pan and Dias [17], the stability analysis of the horseshoe tunnel face was conducted under the linear Mohr–Coulomb failure criterion. Note that the studies listed above are all limited to the linear Mohr–Coulomb failure criterion, but some recent studies [18,19,20] have shown that geotechnical failure often exhibits nonlinear failure characteristics. To solve this shortcoming, various nonlinear failure criterions have been introduced by some scholars to study the stability of geotechnical structures [18,19,20]. However, no attentions have been focused on the stability analysis of the horseshoe tunnel face under nonlinear failure criterion. This study will propose an analytical model to fill this research gap. Compared with previous analytical models, the proposed model has the following two advantages: (1) the whole horseshoe tunnel face is considered in the construction of the analytical failure model in rock masses; and (2) the nonlinear Hoek–Brown failure criterion is incorporated into the stability analysis of the horseshoe tunnel face in rock masses.

This study is organized as follows: (1) a rotational failure model for the horseshoe tunnel face is developed based on Mollon et al. [6]; (2) in the framework of the upper-bound limit analysis method, the limit support pressure is determined under the nonlinear Hoek–Brown failure criterion; (3) the proposed method is validated by comparisons with the numerical results; and (4) the influences of the rock properties on the limit support pressure and the 3D failure surface are presented and discussed.

## 2. Nonlinear Hoek–Brown Failure Criterion

The generalized nonlinear Hoek–Brown failure criterion was first introduced by Hoek [18] to analyze the stability of engineering structures in rock masses and now has been widely adopted to analyze stability issues [18,19]. The generalized nonlinear Hoek–Brown failure criterion can be described as follows:(1)$$\sigma_{1}=\sigma_{3}+\sigma_{ci}{\left(m\frac{\sigma_{3}}{\sigma_{1}}+s\right)}^{a}$$
where $\sigma_{1}$ and $\sigma_{3}$ denote the maximum and minimum principal stresses, respectively, and $\sigma_{ci}$ denotes the uniaxial compressive strength of the rock material. The values s, a and m can be obtained by Equations (2)–(4):(2)$$m=m_{i}\exp\left(\frac{GSI-100}{28-14D_{i}}\right)$$
(3)$$s=\exp\left(\frac{GSI-100}{9-3D_{i}}\right)$$
(4)$$a=\frac{1}{2}+\frac{1}{6}\left[\exp\left(-\frac{GSI}{15}\right)-\exp\left(-\frac{20}{3}\right)\right]$$
where $m_{i}$ is the nonlinear Hoek–Brown coefficient; GSI is the geological strength index; and *D_i_* is the disturbance coefficient of rock.

According to Equations (1)–(4), there are four input parameters for the generalized nonlinear Hoek–Brown failure criterion, namely $\sigma_{ci}$, GSI, $m_{i}$, and *D_i_*. The values of these four parameters can be obtained from Prise [19]. In order to estimate the stability of geotechnical structures using the generalized nonlinear Hoek–Brown failure criterion, Yang et al. [20] proposed a tangent technique to transform the nonlinear failure criterion into a linear one, as shown in Figure 1. From Figure 1, it can be seen that the tangent technique can provide upper-bound solutions for stability issues because the tangent line is above the nonlinear envelope. In Figure 1, the tangent line at point M can be expressed as follows:(5)$$\tau =c_{t}+\sigma_{n}\tan\varphi_{t}$$
where *τ* and *σ_n_* denote the shear stress and normal stress at the failure surface, respectively; $c_{t}$ and $\varphi_{t}$ are the equivalent cohesion and equivalent friction angle, respectively; and $c_{t}$ can be calculated as follows:(6)$$\frac{c_{t}}{\sigma_{ci}}=\frac{\cos\varphi_{t}}{2}{\left[\frac{ma\left(1-\sin\varphi_{t}\right)}{2\sin\varphi_{t}}\right]}^{\frac{a}{1-a}}-\frac{\tan\varphi_{t}}{m}\left(1+\frac{\sin\varphi_{t}}{a}\right){\left[\frac{ma\left(1-\sin\varphi_{t}\right)}{2\sin\varphi_{t}}\right]}^{\frac{1}{1-a}}+\frac{s}{m}\tan\varphi_{t}$$

It is worth noting that the equivalent cohesion $c_{t}$ is the mutative parameter with the change in equivalent friction angle $\varphi_{t}$. In this study, the equivalent friction angle $\varphi_{t}$ is not a given value but an additional optimization parameter, which is different from the linear Mohr–Coulomb failure criterion.

## 3. Rotational Failure Model for the Horseshoe Tunnel Face

Figure 2 shows the rotational failure model for the horseshoe tunnel face. From Figure 2, it can be seen that the proposed failure model rotates around a horizontal *x*-axis passing through point *O* with a uniform angular velocity *ω*. As shown in Figure 2, the failure model proposed in this study can be divided into two parts, S1 and S2. The proposed failure model is assumed to be bounded by two discrete log-spirals as follows:(7)$$\{\begin{array}{c} r_{i}=r_{A}e^{\left(\beta_{i}-\beta_{A}\right)\tan\varphi }\\ r_{j}=r_{B}e^{\left(\beta_{B}-\beta_{j}\right)\tan\varphi } \end{array}$$
where $r_{A}$ and $r_{B}$ are the rotation radius of points A and B, respectively; and $\beta_{A}$ and $\beta_{B}$ are rotation angles of points A and B, respectively. Based on the discretization and the “point-by-point” techniques [6], the proposed failure model can be constructed by following five steps: (1) based on the discretization technique, the horseshoe tunnel face is first uniformly discretized into n points, namely points $A_{1},A_{2},A_{3}\ldots A_{n/2}$ and $B_{1},B_{2},B_{3}\ldots B_{n/2}$ (see Figure 2); (2) the failure part S1 is discretized into $n_{1}$ parts by a series of planes passing through the horseshoe tunnel face and the rotation center *O* (see Figure 2); (3) the failure part S2 is discretized into $n_{2}$ parts by a series of planes with angle dβ between adjacent planes passing through point O; (4) based on the “point-by-point” technique, the point $P_{i,j+1}$(i=1,2,3,…N, j=1,2,3,…N) at the failure surface is generated from the given points *P*_*i*,*j*_ and *P*_*i*+1,*j*_ with the starting points $A_{1},A_{2},A_{3}\ldots A_{n/2}$ and $B_{1},B_{2},B_{3}\ldots B_{n/2}$ at the horseshoe tunnel face (see Figure 3); and (5) the 3D failure surface for the horseshoe tunnel face can be obtained by connecting all adjacent points, $P_{i,j}$, $P_{i+1,j}$ and $P_{i,j+1}$, as shown in Figure 4. More details about the discretization and the “point-by-point” techniques can be found in Mollon et al. [6].

## 4. Work Rate Calculations

Based on the proposed failure model, the limit support pressure of the tunnel face can be derived by equating the work rate exerted by external force to the internal energy dissipation rate using the upper-bound limit analysis method [21,22,23,24,25], which described by Equation (8). Note that the external forces in this study mainly include the supporting pressure at the tunnel face and the rock gravity.
(8)$${\dot{W}}_{T}+{\dot{W}}_{D}={\dot{W}}_{\gamma }$$
where ${\dot{W}}_{T}$ denotes the work rate exerted by the supporting pressure; ${\dot{W}}_{D}$ denotes the internal energy dissipation rate; ${\dot{W}}_{\gamma }$ denotes the work rate exerted by the rock gravity. The details of the work rate calculation will be presented in following subsections.

### 4.1. Internal Energy Dissipation Rate

As shown in Figure 5, the internal energy dissipation rate of the proposed failure model can be expressed by:(9)$${\dot{W}}_{D}=\iint_{s}c_{t}v\cos\varphi_{t}ds=\sum_{i=1}^{N}\sum_{j=1}^{N}\left(c_{t}\omega \cos\varphi_{t}R_{i,j}S_{i,j}\right)$$
where $c_{t}$ is the equivalent cohesion, which can be derived by Equation (6); $R_{i,j}$ is the rotation radius of the center of the discretization surface $F_{i,j}$; and $S_{i,j}$ is the area of the discretization surface $F_{i,j}$, as shown in Figure 5.

### 4.2. Work Rate Exerted by the Rock Weight

The work rate of the rock weight can be calculated by:(10)$${\dot{W}}_{\gamma }={∭}_{V}\gamma vdV=\omega \gamma \sum_{i=1}^{N}\sum_{j=1}^{N}\left(R_{i,j}V_{i,j}\sin\beta_{i,j}\right)$$
where $V_{i,j}$ is the volume of each discretization block of the proposed failure model and $\beta_{i,j}$ is the rotation angle of the discretization block $V_{i,j}$ corresponding to the discretization surface $F_{i,j}$ (see Figure 5).

### 4.3. Work Rate Exerted by the Supporting Pressure

The work rate of the supporting pressure acting on the horseshoe tunnel face can be calculated by:(11)$${\dot{W}}_{T}=\iint_{\Sigma }\sigma_{T}vd\Sigma =\sum_{i}\left(\Sigma_{i}R_{i,0}\cos\beta_{i,0}\right)$$
where $R_{i,0}$ is the rotation radius of the center of the discretization surface $F_{i}$ and $\beta_{i,0}$ is the rotation angle of the center of the discretization surface $F_{i}$.

### 4.4. Determination of the Limit Support Pressure

Combining Equation (8) up to Equation (11), the limit support pressure can be calculated by:(12)$$\sigma_{T}=\gamma HN_{\gamma }-c_{t}N_{c_{t}}$$
where $N_{\gamma }$ and $N_{c_{t}}$ are the nondimensional coefficients, representing the influence of the rock gravity and equivalent cohesion on the limit support pressure. With the combination of Equations (8)–(11), the expressions of $N_{\gamma }$ and $N_{c_{t}}$ can be written as:(13)$$N_{\gamma }=\frac{\sum_{i=1}^{N}\sum_{j=1}^{N}\left(R_{i,j}V_{i,j}\sin\beta_{i,j}\right)}{H\sum_{i}\left(\Sigma_{i}R_{i,0}\cos\beta_{i,0}\right)}$$
(14)$$N_{c_{t}}=\frac{\sum_{i=1}^{N}\sum_{j=1}^{N}\left(c_{t}\omega \cos\varphi_{t}R_{i,j}S_{i,j}\right)}{\sum_{i}\left(\Sigma_{i}R_{i,0}\cos\beta_{i,0}\right)}$$

Combining Equation (12), a nonlinear minimum optimization algorithm named fminsearch is adopted to calculate the limit support pressure with respect to $\beta_{E}$, $r_{E}/H$, and $\varphi_{t}$ in this study. To ensure a responsible failure model of the tunnel face, the optimization for the limit support pressure is under the constraint expressed in Equation (15).
(15)$$\{\begin{array}{c} 0<\beta_{E}<\frac{\pi }{2}\\ \frac{1}{2}<\frac{r_{E}}{H}<20\\ 0<\varphi_{t}<\frac{\pi }{4} \end{array}$$

## 5. Validation

Since no analytical models have been proposed to investigate the face stability of the horseshoe tunnel in rock masses, the numerical results obtained from the numerical software FLAC^3D^ were employed to validate the proposed method. Figure 6 shows the established numerical model based on the FLAC^3D^. As a symmetrical tunnel is considered, the assessment of the tunnel face stability using the numerical approach is only based on half of the horseshoe tunnel (see Figure 6). It is worth noting that the numerical model is large enough (the length, width, and height of the numerical model, respectively, equaling 130 m, 100 m, and 60 m) to eliminate the effect of boundary conditions. As the rock deformation occurs mainly near the tunnel face, the mesh in the vicinity of the tunnel face is densified in order to improve the calculation efficiency (see Figure 6). As also shown in Figure 6, the boundary conditions of the numerical model are as follows [3,4]: the bottom of the numerical model is full-fixed, the vertical faces of the numerical model are fixed in the normal directions, and the top of the numerical model is free to displace. An elastic–plastic constitutive model following the nonlinear The Hoek–Brown failure criterion is adopted to simulate the rock masses, and the shell structure is used to simulate the tunnel lining element with the thickness being 22 mm, Poisson’s ratio being 0.2, and Young’s modulus being 15 Gpa. The rock properties adopted in the comparisons are as follows: $\gamma =25\;{\mathrm{kN}/m}^{3}$, $m_{i}=5-35$, GSI=20−30, $\sigma_{ci}=1\;\mathrm{MPa}$, and $D_{i}=0.5$ [19]. The Young’s Modulus *E_m_* and passion ratio *V_m_* of the rock mass can be obtained by [26,27]:(16)$$\{\begin{array}{c} E_{m}=10^{5}\cdot \left\{\frac{1-0.5\cdot D_{i}}{1+\exp\left[\left(75+25\cdot D_{i}-GSI\right)/11\right]}\right\}\\ v_{m}=-0.002\cdot GSI-0.003\cdot m_{i}+0.457 \end{array}$$

Based on the established numerical model, by gradually reducing the supporting pressure until the tunnel face is in a critical state, the corresponding supporting pressure can be recognized as the limit support pressure *σ_T_* [28]. A comparison between the limit support pressure obtained by the proposed method and the results from the numerical model is presented in Figure 7. As shown in Figure 7, the value of limit support pressure obtained by the proposed method greatly decreases with the increasing of $m_{i}$, which is consistent with the numerical results. Additionally, the limit support pressures obtained in this study are slightly lower than the numerical results, with a maximum difference less than 8%. This observation illustrates that the proposed method can be used to accurately estimate the stability of the horseshoe tunnel face under the nonlinear Hoek–Brown failure criterion.

Furthermore, the numerical simulation method is time-consuming and usually requires some redundant input variables, some of which are difficult to specify [28]. Compared with the numerical simulation method, the proposed method is easier to implement and can provide a vehicle for engineers to directly design required limit support pressures for the horseshoe tunnel face. In summary, one can conclude that the proposed method is an efficient and accurate approach to determining the face stability of the horseshoe tunnel in rock masses.

## 6. Parametric Analysis

### 6.1. Effect of the Geological Strength Index, GSI, on the Tunnel Face Stability

In this section, the effect of the geological strength index, GSI, on the limit support pressure of the horseshoe tunnel face is discussed (see Figure 8). The curve of normalized limit support pressure, *σ**_Τ_**/**γ**H*, with GSI is presented in Figure 8. The adopted parameters are L=11.6 m, H=10.3 m, $\gamma =25\;{\mathrm{kN}/m}^{3}$, $\sigma_{ci}=1\;\mathrm{MPa}$, $D_{i}=0.5$, $m_{i}=5–35$, and GSI=10–60. It can be seen in Figure 8 that the normalized limit support pressure, *σ_T_*/*γH*, decreases with the geological strength index, GSI, increases, and the decreasing rate decreases with the increasing of GSI. For instance, for $m_{i}=10$, the normalized limit support pressure, *σ_T_*/*γH*, decreases from 0.35 to 0.03 when GSI increases from 10 to 50. This indicates that a lower limit support pressure is associated with a high value of GSI, and a high *GSI* is beneficial for the tunnel face stability.

Figure 9 presents the effect of the geological strength index, GSI, on the 3D failure surface for L=11.6 m, H=10.3 m, $\gamma =25\;{\mathrm{kN}/m}^{3}$, $\sigma_{ci}=1\;\mathrm{MPa}$, $D_{i}=0.5$, $m_{i}=5$, and GSI=20–50. It can be seen in Figure 9 that the 3D failure surface is large for high values of *GSI*. Combining this with Figure 8, one can conclude that a larger 3D failure surface is associated with a high value of the normalized limit support pressure.

### 6.2. Effect of the Hoek–Brown Coefficient $m_{i}$ on the Tunnel Face Stability

Figure 10 shows the curve of the normalized limit support pressure *σ_T_*/*γH* with the Hoek–Brown coefficient $m_{i}$. The analytical parameters are L=11.6 m, H=10.3 m, $\gamma =25\;{\mathrm{kN}/m}^{3}$, $\sigma_{ci}=1\;\mathrm{MPa}$, $D_{i}=0.5$, GSI=10–50, and $m_{i}=5–35$. Figure 11 presents the effect of the Hoek–Brown coefficient, $m_{i}$, on the 3D failure surface for L=11.6 m, H=10.3 m, $\gamma =25\;{\mathrm{kN}/m}^{3}$, $\sigma_{ci}=1\;\mathrm{MPa}$, $D_{i}=0.5$, GSI=10, and $m_{i}=10–35$. As shown in Figure 10, the normalized limit support pressure, *σ_T_*/*γH*, decreases with the Hoek–Brown coefficient, $m_{i}$, increases, and the decreasing rate decreases significantly with the increasing of $m_{i}$. For example, for GSI=10, the normalized limit support pressure *σ_T_*/*γH* decreases from 0.35 to 0.11 when $m_{i}$ increases from 10 to 35. This observation can be illustrated by the change of the 3D failure surface from a larger size to a smaller one with the increasing of the Hoek–Brown coefficient, $m_{i}$, from 10 to 35 (see Figure 11).

### 6.3. Effect of the Uniaxial Compressive Strength, $\sigma_{ci}$, on the Tunnel Face Stability

The effect of the uniaxial compressive strength, $\sigma_{ci}$, on the normalized limit support pressure, *σ_T_*/*γH*, is discussed in this section, and the curve of the normalized limit support pressure, *σ_T_*/*γH*, with different uniaxial compressive strength, $\sigma_{ci}$, values is presented in Figure 12. The adopted parameters are as follows: L=11.6 m, H=10.3 m, $\gamma =25\;{\mathrm{kN}/m}^{3}$, GSI=10, $m_{i}=5$, $D_{i}=0–1.0$, and $\sigma_{ci}=1–10\;\mathrm{MPa}$. Figure 13 presents the effect of the uniaxial compressive strengths, $\sigma_{ci}$, on the 3D failure surface for L=11.6 m, H=10.3 m, $\gamma =25\;{\mathrm{kN}/m}^{3}$, GSI=10, $m_{i}=5$, $D_{i}=0.4$, and $\sigma_{ci}=2–10\;\mathrm{MPa}$. From Figure 10, it can be seen that the normalized limit support pressure, *σ_T_*/*γH*, decreases with the $\sigma_{ci}$ increases. For instance, for $D_{i}=0.4$, the normalized limit support pressure, *σ_T_*/*γH*, decreases from 0.33 to 0.11 when $\sigma_{ci}$ increases from 2 MPa to 10 MPa. This is consistent with Senent et al. [29] for the stability analysis of the circular tunnel face. This observation is illustrated by the changes of the 3D failure surface from a larger size to a smaller one with the increasing of the uniaxial compressive strength, *σ_ci_*, from 2 MPa to 10 MPa (see Figure 13).

### 6.4. Effect of the Disturbance Coefficient of Rock, $D_{i}$, on the Tunnel Face Stability

This section discusses the effect of the disturbance coefficient of rock, $D_{i}$, on the normalized limit support pressure, *σ_T_*/*γH*, and the curve of the normalized limit support pressure, *σ_T_*/*γH*, with different rock disturbance coefficients, $D_{i}$, is presented in Figure 14. The analytical parameters are as follows: L=11.6 m, H=10.3 m, $\gamma =25\;{\mathrm{kN}/m}^{3}$, $m_{i}=5$, GSI=10, $\sigma_{ci}=1–10\;\mathrm{MPa}$, and $D_{i}=0–1.0$. Figure 15 presents the effect of the disturbance coefficient of rock, $D_{i}$, on the 3D failure surface for L=11.6 m, H=10.3 m, $\gamma =25\;{\mathrm{kN}/m}^{3}$, $m_{i}=5$, GSI=10, $\sigma_{ci}=10\;\mathrm{MPa}$, and $D_{i}=0.2–0.8$. It can be seen that the normalized limit support pressure, *σ_T_*/*γH*, increases with the increasing of the rock disturbance coefficient, $D_{i}$, and the increasing rate of the normalized limit support pressure, *σ_T_*/*γH*, increases significantly with the rock disturbance coefficient, $D_{i}$, increases. For instance, for $\sigma_{ci}=10\;\mathrm{MPa}$, the normalized limit support pressure, *σ_T_*/*γH*, increases from 0.07 to 0.40 when $D_{i}$ increases from 0.2 to 0.8. This observation can be illustrated by the changes of the 3D failure surface from a smaller size to a larger one with the increasing of the rock disturbance coefficient, $D_{i}$, from 0.2 to 0.8 (see Figure 15).

## 7. Conclusions

In the framework of the upper-bound limit analysis theory, this study analyzes the stability of horseshoe tunnel faces in rock masses. Compared with previous analytical solutions, the following improvements have been achieved:(1)In this study, the nonlinear Hoek–Brown failure criterion is first incorporated into the stability analysis of horseshoe tunnel faces in rock masses. The comparisons between the results from the proposed method with the numerical results illustrate that the proposed method is an efficient and accurate approach to assessing the face stability of horseshoe tunnels.(2)Based on the proposed method, the effect of rock properties on the normalized limit support pressure and the 3D failure surface are presented. It is shown that, for selected cases, the normalized limit support pressure of the tunnel face greatly decreases with the increasing of *GSI*, *m_i_*, and *σ_ci_*, and decreasing of *D_i_*; the 3D failure surface become larger with the decreasing of *GSI*, *m_i_*, and *σ_ci_*, and increasing of *D_i_*; a larger 3D failure surface is associated with a high value of the normalized limit support pressure; and high *GSI*, *m_i_*, and *σ_ci_* are beneficial for the tunnel face stability.

A possible extension of this work could be further investigation into the active and passive stability of deep/shallow horseshoe tunnel faces [30] considering the seismic loads [31] and the seepage forces [32].

## Figures and Tables

**Figure 1 materials-15-04306-f001:**
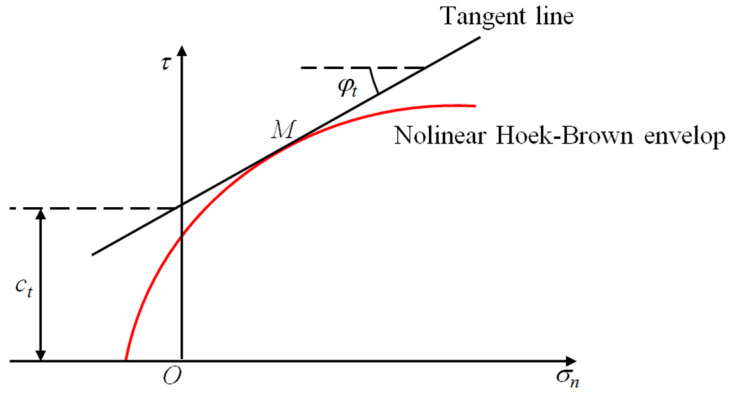
Generalized nonlinear Hoek–Brown failure criterion and tangent technique.

**Figure 2 materials-15-04306-f002:**
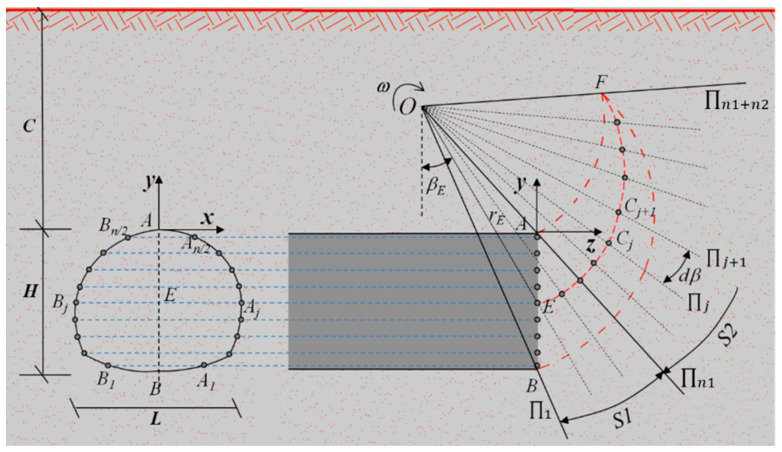
Scheme of the proposed failure model.

**Figure 3 materials-15-04306-f003:**
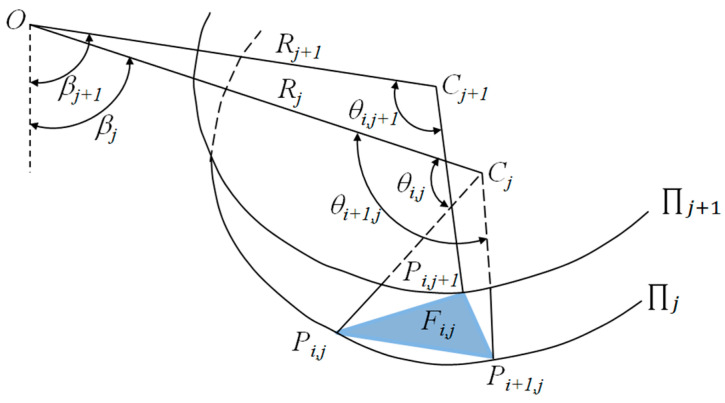
“Point-by-point” technique.

**Figure 4 materials-15-04306-f004:**
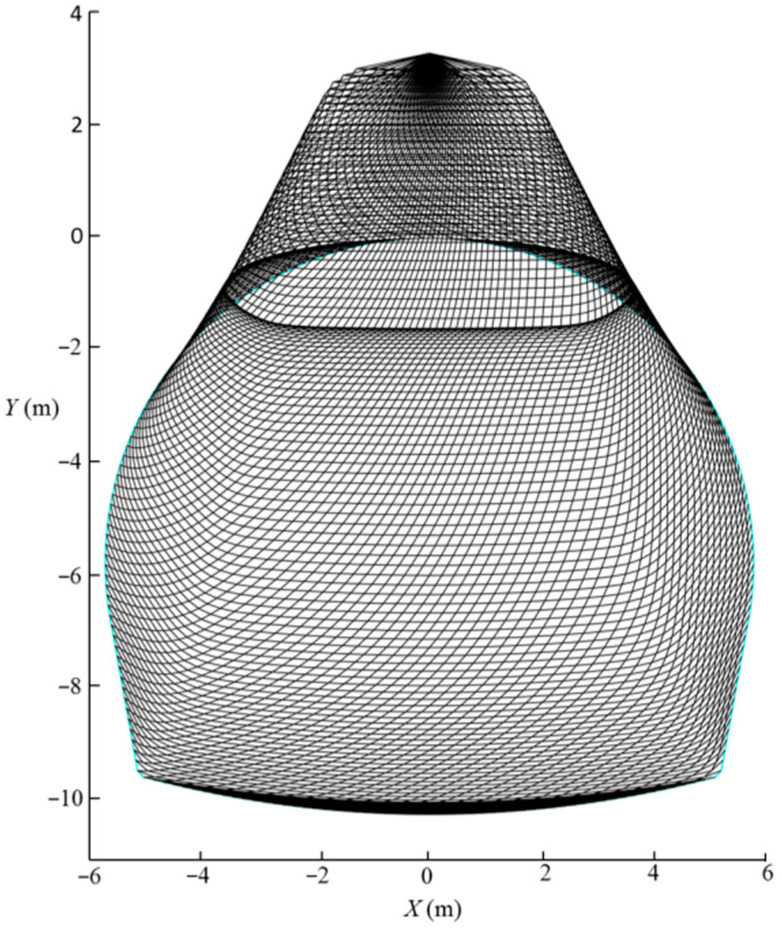
3D failure surface.

**Figure 5 materials-15-04306-f005:**
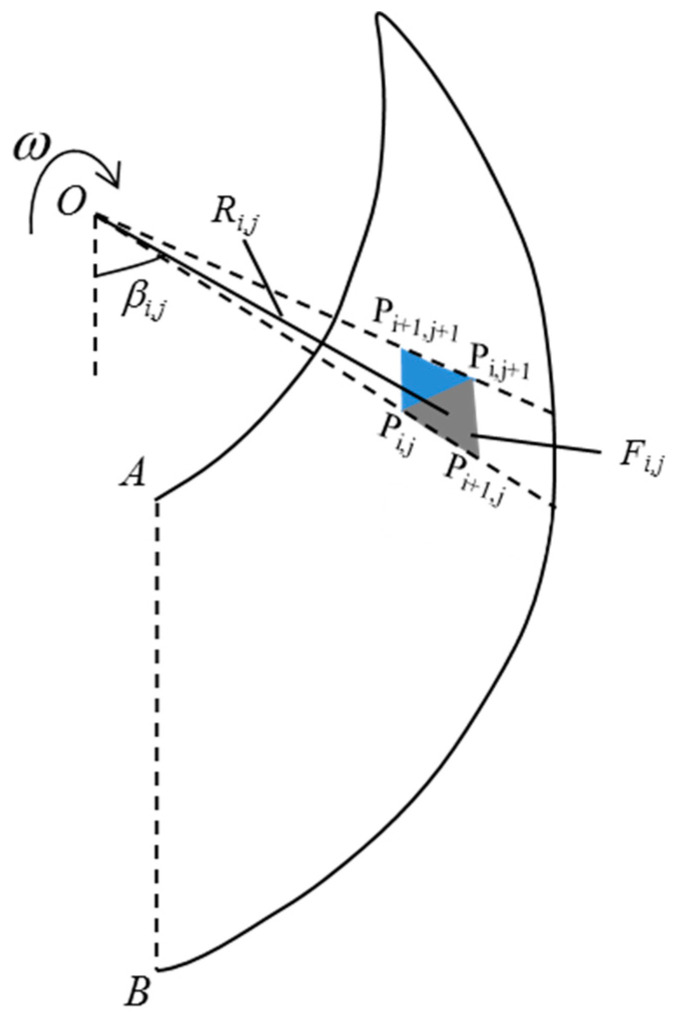
Scheme of the calculation of the internal energy dissipation rate.

**Figure 6 materials-15-04306-f006:**
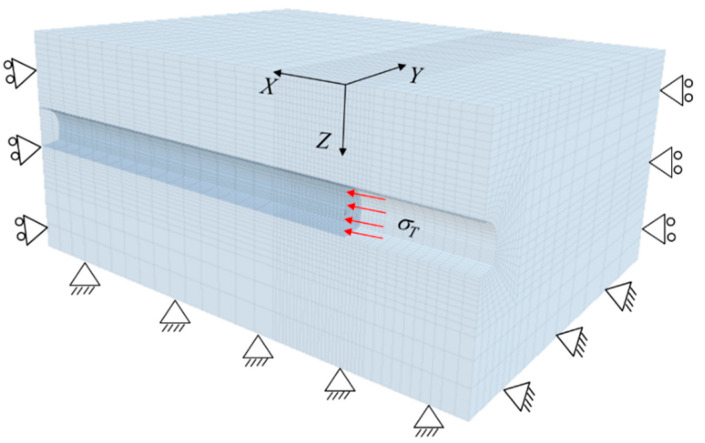
Numerical model.

**Figure 7 materials-15-04306-f007:**
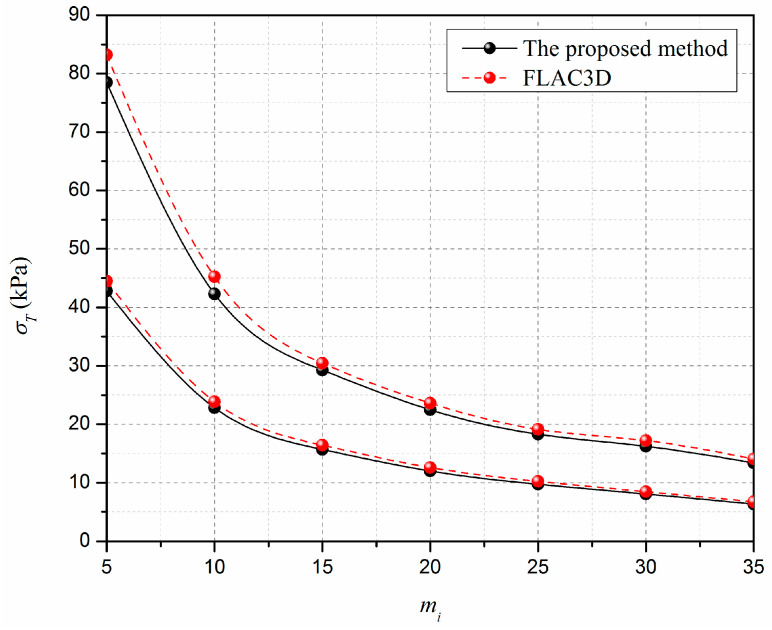
Validation of the proposed method.

**Figure 8 materials-15-04306-f008:**
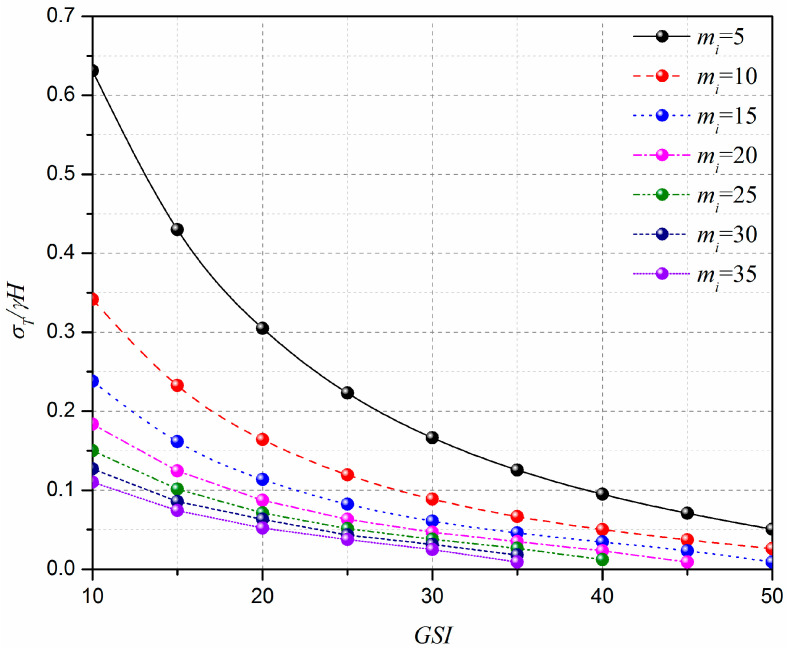
Effect of GSI on the normalized limit support pressure, *σ_T_*/*γH*.

**Figure 9 materials-15-04306-f009:**
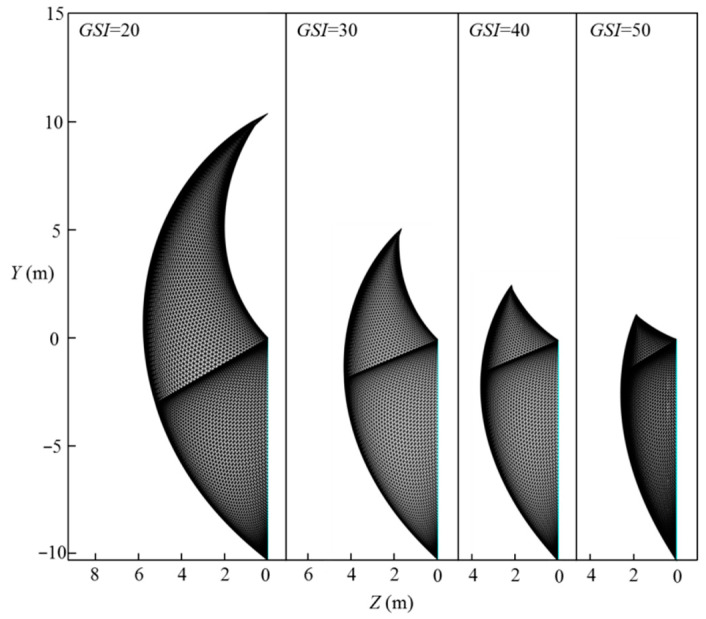
Effect of *GSI* on the 3D failure surface.

**Figure 10 materials-15-04306-f010:**
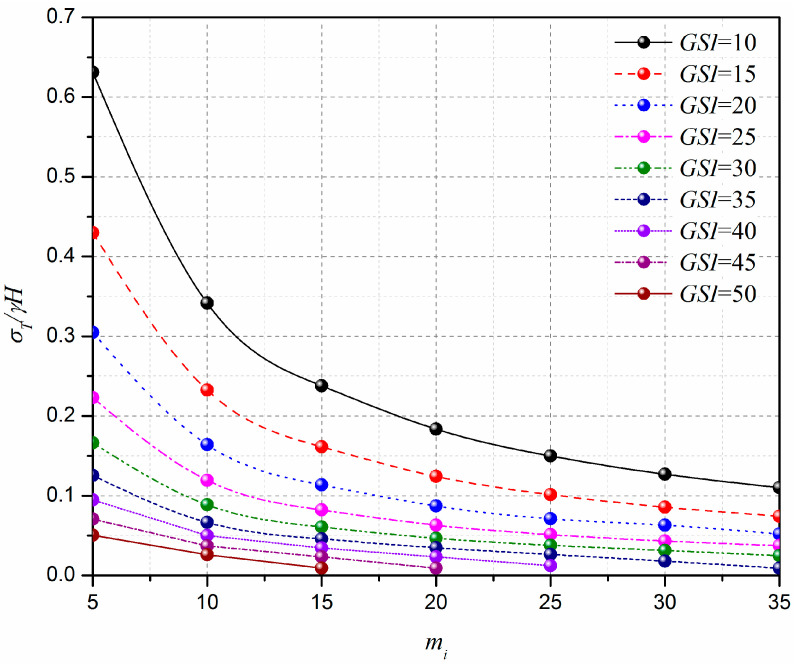
Effect of $m_{i}$ on the normalized limit support pressure *σ_T_/γH*.

**Figure 11 materials-15-04306-f011:**
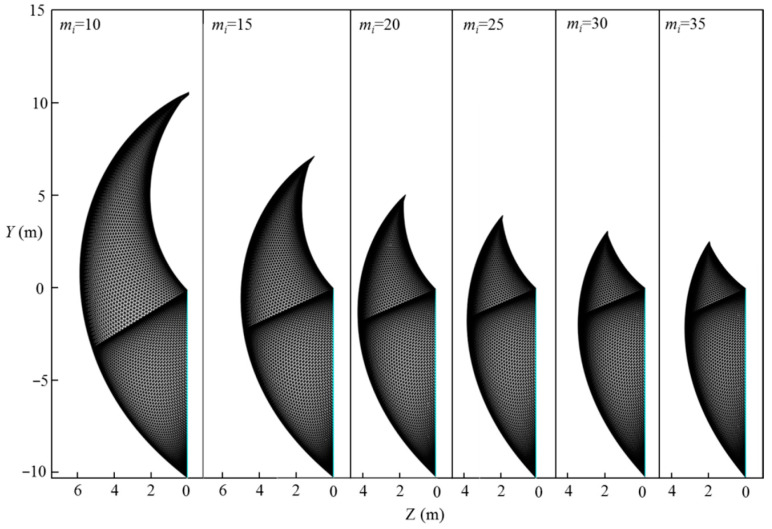
Effect of $m_{i}$ on the 3D failure surface.

**Figure 12 materials-15-04306-f012:**
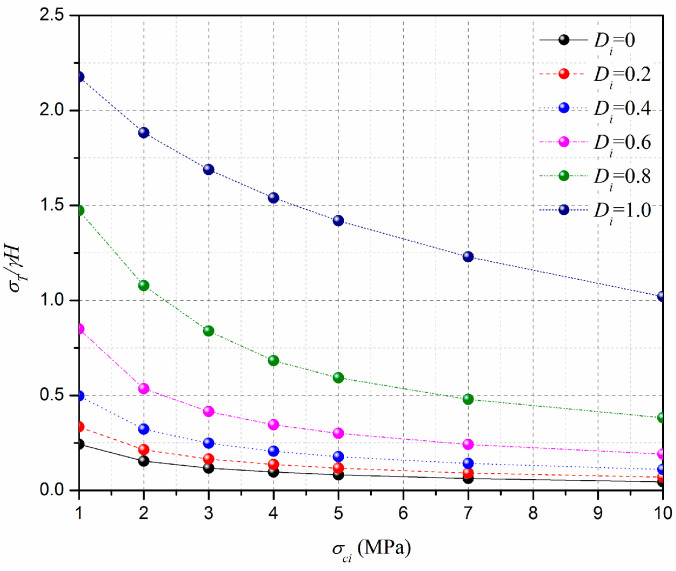
Effect of $\sigma_{ci}$ on the normalized limit support pressure, *σ_T_/γH*.

**Figure 13 materials-15-04306-f013:**
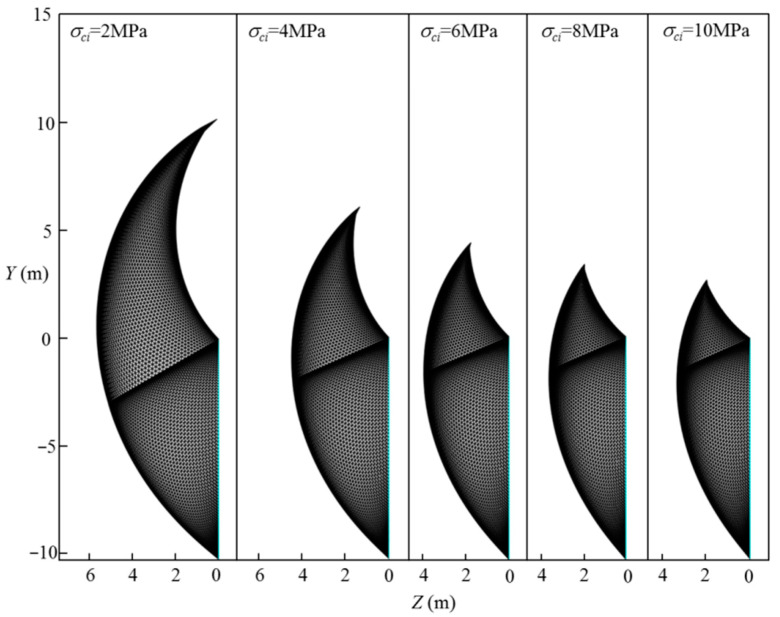
Effect of $\sigma_{ci}$ on the 3D failure surface.

**Figure 14 materials-15-04306-f014:**
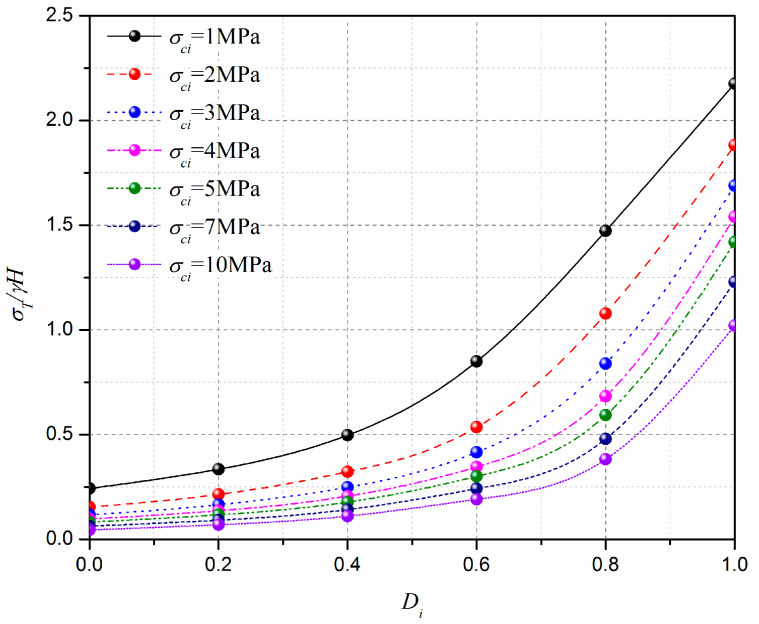
Effect of $D_{i}$ on normalized limit support pressure, *σ_T_/γH*.

**Figure 15 materials-15-04306-f015:**
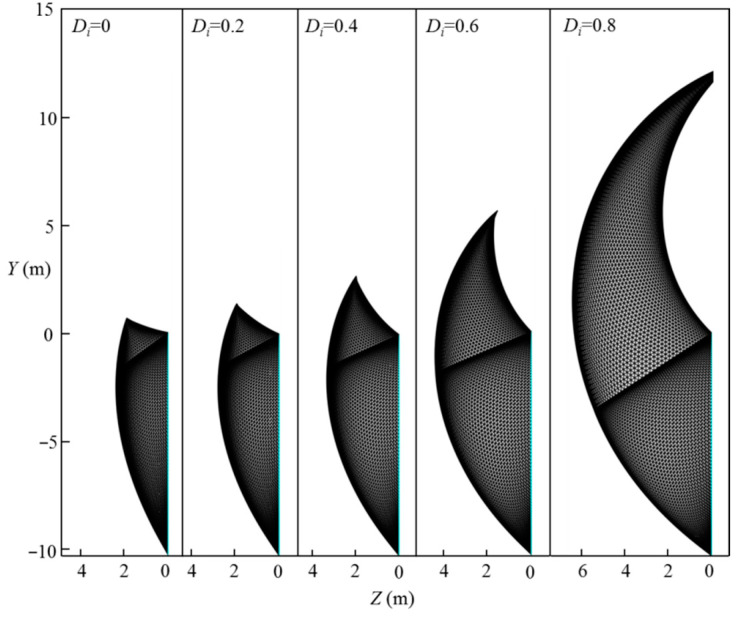
Effect of $D_{i}$ on the 3D failure surface.

## Data Availability

The data presented in this study are available on request from the corresponding author.

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
