# Peer review of "Stability Analysis of the Horseshoe Tunnel Face in Rock Masses"

_materials, 2022, doi:10.3390/ma15124306_

Round 1

Reviewer 1 Report

The authors present a novel model based on the Hoek-Brown failure criterion for horseshoe tunnel face stability analysis. This method is validated with respect to a numerical model in FLAC3D, a well-known simulation software for geotechnical analysis. Once the model is validated, the authors used it to analyze the effect of its four parameters on tunnel stability. The paper is interesting and well structured, although there are some issues that should be addressed before considering it for publication:

1. In the abstract, the model parameters (symbols) are directly used with no definitions. Please explain what GSI, mi, etc... are.

2. The manuscript should undergo an English revision. For instance, in the first paragraph of the Introduction, the term "tunnel" is used 14 times!

3. The introduction misses the research significance. In other words, the authors should stress more (and better) why this contribution is important. Moreover, if the can make an estimate on the economic side, even better. What are the economic advantages of using this approach? Is it better for avoiding disasters? 

4. Line 66: remove the term "effective" from the principal stresses.

5. Equation 5: Shear stress theta is not defined elsewhere.

6. The first paragraph in Section 3 could be placed in the introduction... 

7. Figure 3: There's a term in Chinese.

8. Section 4.2. When referring to "Work rate", shouldn't you be using W dot instead of simply W?

9. Line 183: what do you mean by "encrypted"? Should it be "constrained" instead?

10. Line 192: please cite the source of the values of model parameters.

11. Figure 6: Why such choice of boundary conditions? The horizontal plane, shouldn't it be simply supported?

12. Conclusions: What future work could be done? Please highlight what your model is useful for.

Reviewer 2 Report

This article investigated the stability analysis of the horseshoe tunnel face using Hoek-Brown failure criterion.

1.     In the abstract, it is better to start with current issues in this research and what is the solution required to overcome the issue.

2.     The introduction section can be strengthened by adding more literatures.

3.     What is a point by point technology and how can it be used for the failure model.

4.     The earlier published articles can support the effect of uniaxial compressive stress on tunnel face stability.

5.     The discussion sections are adequate and equations are presented clearly.

6.     How does the accuracy of the proposed is ensured?

7.     The conclusion can be improved by adding key findings.
